# Transdermal Fentanyl Uptake at Two Different Patch Locations in Swiss White Alpine Sheep

**DOI:** 10.3390/ani10091675

**Published:** 2020-09-17

**Authors:** Tim Buchholz, Maria Hildebrand, Anja Heider, Valentina Stenger, Daniel Arens, Claudia Spadavecchia, Stephan Zeiter

**Affiliations:** 1AO Research Institute Davos, 7270 Davos Platz, Switzerland; tim.buchholz@aofoundation.org (T.B.); maria.hildebrand@aofoundation.org (M.H.); valentina.stenger@aofoundation.org (V.S.); daniel.arens@aofoundation.org (D.A.); 2Swiss Institute of Allergy and Asthma Research (SIAF), University Zurich, 7265 Davos Wolfgang, Switzerland; anja.heider@siaf.uzh.ch; 3Vetsuisse Faculty, Department for Veterinary Medicine, University of Bern, 3012 Bern, Switzerland; claudia.spadavecchia@vetsuisse.unibe.ch

**Keywords:** 3R principles, refinement, analgesia, transdermal fentanyl patch, sheep

## Abstract

**Simple Summary:**

Providing adequate and prolonged pain relief to sheep undergoing invasive orthopedic surgery while keeping side effects and stress for the animals at a minimum is challenging. Transdermal patches continuously releasing the synthetic opioid fentanyl through the skin, are a frequently used method in veterinary and human medicine. To refine the current analgesia protocol, we compared the uptake of fentanyl from a transdermal fentanyl patch applied at two different skin locations in sheep. The fentanyl plasma levels were measured at different time points over five days. The patch applied on the foreleg resulted in a faster fentanyl uptake with higher peaks and a longer time within or above the target fentanyl plasma concentration when compared to the one on the thorax. Additionally, it was easier to apply the patch at the foreleg than at the thorax. Our findings suggest that the fentanyl patch should be applied to the foreleg 3–6 h before the painful insult and that its effect should last at least 48 h.

**Abstract:**

When using animals in biomedical research, investigators have the responsibility to ensure adequate analgesia. Currently, transdermal fentanyl patches (TFP) are often used to provide postoperative analgesia in large laboratory animals. The aim of this study was to compare the fentanyl uptake resulting from TFP applied at two different locations, namely the foreleg and the thorax, in healthy adult sheep. Twelve sheep received a TFP with an intended dosage of 2 ug/kg/h. Blood samples were taken at different time points over a period of five days and the fentanyl plasma levels were measured. The TFP applied on the foreleg allowed a faster fentanyl uptake with higher peaks and a longer time within or above the target concentration of 0.6–1.5 ng/mL, shown to be analgesic in humans, when compared to the one on the thorax. Assuming that the effective plasma concentration described for humans is providing analgesia in sheep as well, the present findings suggest that it should be sufficient to apply the TFP 3–6 h before the painful insult and that its effect should last at least 48 h. Furthermore, when TFP are used to provide postoperative analgesia in sheep, they should be placed on the foreleg rather than on the thorax.

## 1. Introduction

Animals are frequently used in research, in particular in biomedical studies [1]. The selection of the appropriate animal species and a model for the specific scientific question under investigation are the first steps towards a successful study [2,3]. If the species and model utilized have not been tailored to the study goal, it might prove impossible to answer the research question. Furthermore, when planning an animal study, it is necessary to consider the 3R principles. An animal study should only be considered if no alternative method exists to answer the research question (replace). Further, by careful planning, it should also be ensured that a maximum of relevant information is obtained from as few animals as possible (reduce), and last but not least, the burden of the animals is alleviated as much as possible by adapting standard protocols and procedures to the chosen species and model (refine) [2].

Sheep are commonly used as a preclinical model in orthopedic research [1,4,5,6,7,8,9]. They are chosen due to their similarities to humans in terms of body weight, rate of bone healing [7], and characteristics of mineralization of long bones [1]. Additionally, the size of several bones can be compared with the one of humans [10]. For example, sheep were employed by Yamamuro et al. to test a prosthesis for the replacement of the lumbar vertebrae [6], and by den Boer et al. to develop a new segmental long bone defect model [7]. In an implant-associated infection study, Boot et al. used a sheep model to investigate different treatment strategies after implantation of a intramedullary implant in the tibia [8]. In all these models and the majority of orthopedic research, surgical intervention is essential to mimic the clinical problem. As in clinical cases, surgical interventions require the implementation of an appropriate analgesic protocol. In research, the analgesic protocol is not only a tool but can also be an important determinant for the study outcome since pain can majorly impact the reproducibility of results [11]. Therefore, an analgesic protocol tailored to the species and model is a necessity not only for animal welfare reasons, but also to ensure the acquisition of high-quality data.

In order to control pain produced by invasive surgeries, analgesic drugs are typically administered pre-, intra- and a postoperatively. Before and during surgery, the sheep is under general anesthesia with a venous access allowing easy administration of analgesic medication intravenously. On the contrary, after surgery, the sheep is awake, and administration of pain medication can be stressful. In addition, pain assessment during the postoperative phase is very challenging and insufficient analgesia could easily be unrecognized [12,13]. An adequate postoperative analgesic protocol should therefore guarantee constant analgesia adapted to the level of pain inflicted with minimal side effects and stress.

Opioids are analgesics for moderate to severe pain, often used to obtain sufficient pain relief due to invasive surgical procedures [14]. Fentanyl is a strong synthetic opioid commonly used in analgesia protocols for humans and animals [15,16,17,18,19]. Due to the short half-life of this drug, systems allowing continuous administration are required [16]. During surgery, fentanyl can be given by continuous rate infusion (CRI) via infusion pumps. Such tools can be heavy and can disturb the freedom of movement of sheep when awake. For this reason, fentanyl, as an injectable medication, is not the method of choice during the postoperative phase. However, fentanyl can also be applied via a transdermal fentanyl patch (TFP), which consists of a polyacrylate adhesive layer allowing absorption of fentanyl through the skin (according to the information provided by Mepha^®^ Pharma AG). With such a TFP providing a continuous and prolonged fentanyl administration, the sheep can be left undisturbed, neither stressful injections nor infusion pumps are needed [20,21]. Therefore, the transdermal application of fentanyl is a suitable method to provide analgesia for a time period of several days [16].

In sheep, the foreleg is a commonly used location for TFP since the application is very simple [20,22]. However, not only the ease of application but also the onset of action as well as the overall achieved fentanyl plasma concentration should be considered. In a study by Ahern et al., the TFP was applied 12 h prior to surgical intervention to compensate the slow uptake and achieve analgesic levels intraoperatively [22]. In dogs, TFP application has been recommended even 24 h before surgery [23]. In horses however, it has been shown that the application of a TFP at the foreleg results in a lower absorption rate and lower fentanyl plasma levels when compared to the application at the groin region or thorax [24]. In another study in horses by Orsini et al., the TFP applied at the thorax showed a fast fentanyl uptake [25]. Considering these findings in horses, it is logical to hypothesize that the thorax might be preferable in sheep as well, because of better skin perfusion, less affected by temperature variation and lower mobility when compared to the limb. Additionally, the onset of action might be faster, making it unnecessary to apply the TFP many hours before its desired effect.

The comparison of different locations for TFP application in sheep has not been described yet.

The aim of this study was to compare the fentanyl uptake resulting from a TFP applied at two different locations, namely the foreleg and the thorax, in healthy adult sheep.

Following TFP application, fentanyl plasma levels were measured at different time points. It was hypothesized that with the TFP applied at the thorax the fentanyl uptake would be faster and a higher peak would be reached than with the TFP applied at the foreleg.

## 2. Materials and Methods

### 2.1. Study Design

This experimental, prospective, randomized study was approved by the Cantonal Committee for Animal Experiments of Graubünden (GR-TVB No. 35_2018) and conducted in an AAALAC International approved facility. The sheep were randomly assigned to one of two equal groups (*n* = 6 per group) by drawing lots. An ear tag with an internal consecutive number was used as identification. In one group, the TFP (fentanyl Mepha^®^ Matrix patches/Mepha Pharma AG/57362) was applied to the foreleg and in the other group to the thorax (Figure 1)

The foreleg was defined as the middle of the lateral surface starting from the elbow joint proximally and ending at carpal joint distally. The region of the thorax was defined as approx. 5 cm caudal from the scapula, at the thorax (Figure 2). The primary outcome, the fentanyl plasma level, was measured at 12 different time points.

### 2.2. Animals

Sheep included in this study were all females and more than 2 years old and determined to be healthy based on clinical examination, hemogram (hematocrit and white blood cell count determined by a “Vet ABC” (ABX Diagnostics, France) and biochemistry analysis (total proteins measured with a refractometer). None of the sheep had undergone anesthesia or surgery before to exclude any prior effects. In total, 13 female White Swiss Alpine sheep were part of this study (mean age 3.2, range 2 to 6.5 years; mean weight 76.2 kg, range 59.5 to 99.5 kg). The sheep were acclimatized to the research facility and the daily routine for 2 weeks in shared pens prior to TFP application. The stable had a 12 h day/night cycle and all sheep had access to daylight in form of a window. Temperature was maintained between 15 and 20 °C, with relative humidity above 30%, and 10–15 air changes per hour were performed. Maintenance diet consisting of a mixture of straw, hay, silage, maize, and salt was fed twice a day. Water was available ad libitum in an automatic water drinker.

### 2.3. Transdermal Fentanyl Patch Application and Long-Term Catheter

A fentanyl dosage of 2 µg/kg/h was targeted based on previous literature [22,26]. For each sheep, the required dose was approximated using the available TFP sizes of 12 µg/h, 25 µg/h, 50 µg/h and 100 µg/h.

The sheep were sedated with an intramuscular injection of detomidine (0.04 mg kg^−1^; Equisedan; Dr. E. Graeub AG, Bern, Switzerland). After 15 min, the left jugular vein was prepared aseptically and a permanent catheter (Certofix^®^ Mono, B.Braun, Sempach, Switzerland) was placed using a Seldinger method [27]. The catheter was sutured with a 2-0 Ethilon II (Ethicon Inc., Somerville, NJ, USA) on three points and covered with adhesive retention dressing. Additionally, a cotton bandage and cohesive bandage was applied to prevent the catheter from loosening. The TFP location was then prepared according to the group. The area was clipped (Figure 2) (foreleg group: 10 cm wide band around the foreleg; thorax group: 15 × 15 cm square on left thorax) and carefully shaved with warm water and soap (Softaskin^®^, B.Braun). Then, the area was degreased by scrubbing three times with Ethanol 80% and allowed to dry for 5 min. The TFP were applied 75 min after the sedation. This time was chosen so all sheep had the same sedation effect when the TFP was applied. The TFP was covered with a self-adhesive “vet polster bandage” by HENRY SCHEIN^®^ (product code: 900-9911). For the foreleg, this bandage was cut to fit once around the foreleg while for the thorax a square was cut out about 2–3 cm larger than the area covered by the TFP. Above the polster bandage, a cohesive bandage (“cohesive elastic crepe bandage”, 20 m × 10 cm, HENRY SCHEIN^®^, product code: 900-8588) was applied. Depending on the TFP location, this bandage was either wrapped around the foreleg or around the entire thorax (Figure 3).

### 2.4. Blood Withdrawal

Blood samples were taken at 12 different time points: 0 (before TFP application), and then 3, 6, 9, 12, 18, 24, 36, 48, 72, 96, and 120 h after the application of the TFP (Figure 1). All blood samples were taken using the long-term catheter. The first 5–6 mL were discarded and afterwards an EDTA blood tube (9 mL S-Monovette^®^, Sarstedt AG&Co.KG) was filled. Then, the catheter was flushed with 10 mL Saline with 500 IE Heparin/mL (Heparinum natricum, FRESENIUS Medical Care (Schweiz) AG). Within two hours after withdrawal, the blood samples were centrifuged for 10 min with 2500 rpm. The plasma was aliquoted in Eppendorf tubes (Vaudaux-Eppendorf AG, Schönenbuch, Switzerland) and frozen at −80 °C until analysis.

### 2.5. Blood Analysis

Fentanyl plasma levels were analyzed by a human enzyme-linked immunoabsorbent assay (Forensic ELISA-Kit, Neogen Toxicology). Standard values were created by adding known fentanyl concentrations (8 in total, fentanyl 0.5 mg/10 mL solution for injection, Sintetica S.A., Switzerland) to plasma of fentanyl-naive sheep. Fentanyl-naïve plasma was used as the blank. The absorbance was read at 450 nm using a Mithras microplate reader (Berthold Technologies). Both standards and samples were measured in duplicates and the assay was performed according to the protocol.

### 2.6. Study Outcomes

The primary outcome of this study was the fentanyl plasma level. As an analgesic fentanyl plasma level for sheep is not described in the literature, the minimum analgesic fentanyl plasma level described for opioid-naive human patients and corresponding to the concentration range of 0.6–1.5 ng/mL was used as a reference in the present study [28]. Time to reach the concentration threshold of 0.6 ng/mL and duration of effect, defined as fentanyl concentration > 0.6 ng/mL, were evaluated.

Secondary outcomes were recorded twice a day from TFP application until the end of the study (5 days) to detect potential fentanyl related side effects. Physiological parameters, including rectal body temperature and respiratory rate, water and food uptake, as well as amount and texture of feces were evaluated at regular intervals. Furthermore, the condition of the bandage and the patch application site were evaluated. The TFP was removed after the last blood sample was taken. Pictures of the skin from every sheep were taken to assess the skin reaction to the TFP or to the bandage. General behavior, and in particular the occurrence of sedation or excitation, was observed and described. Additionally, the body weight was taken before TFP application and after TFP removal, always before sheep were fed. These data were collected by the same observer for all time points. This person was also responsible for handling of the sheep before the study commenced, clinical examination, and the blood work.

### 2.7. Statistics

A descriptive statistical analysis was performed for the data of this study as a statistically significant difference is not necessarily connected to an appropriate fentanyl level. The number of sheep that achieved the predefined fentanyl plasma level of 0.6–1.5 ng/mL was determined. The goal was to define the time period during which the expected minimum fentanyl plasma level was achieved in every individual sheep.

## 3. Results

Thirteen sheep were part of this study. One sheep of the thorax group was excluded as the TFP detached 18 h after placement. Data from this sheep were excluded from the study. All sheep recovered well after sedation.

### 3.1. Primary Outcome: Fentanyl Plasma Level

Based on the selected TFP size, the overall average expected fentanyl administration rate was 2.08 µg/h/kg (range 2.01 to 2.16 µg/h/kg). The average expected fentanyl administration rate was 2.01 µg/h/kg (range 2.01 to 2.16 µg/h/kg) in the foreleg group and 2.07 µg/h/kg (range 2.01 to 2.1 µg/h/kg) in the thorax group.

In the foreleg group, five out of six sheep achieved a fentanyl plasma level threshold of >0.6 ng/mL after 3 h, and all sheep within 6 h. At least this level was maintained in all sheep for a minimum of 48 h and in four out of six sheep for a maximum of 72 h (Table 1, Figure 4). In the thorax group, only two sheep had fentanyl levels above 0.6 ng/mL after 3 h, four sheep after 6 h, five sheep after 9 h, and all six sheep after 12 h (Table 1, Figure 5). At 24 h, all six sheep still had a concentration above 0.6 ng/mL. However, thereafter it started to decline so that, by 36 h, the level of 0.6 ng/mL was achieved in only five sheep, at 72 h in three sheep, and by 96 h two sheep had level above 0.6 ng/mL. When comparing both groups, the fentanyl uptake was more consistent in terms of the onset of the desired plasma concentration as well as duration of effect (Figure 6), when applied to the foreleg than to the thorax. At 120 h, no sheep from either group were above the predefined threshold levels.

All sheep in the foreleg group were above 1.5 ng/mL at least once during the study, with 5/6 sheep reaching this level already after 3 h. In contrast, two out of six sheep of the thorax group never attained a fentanyl plasma level of 1.5 ng/mL and only one sheep reached this value after 3 h. A fentanyl plasma level > 1.5 ng/mL was achieved in all sheep of the foreleg group between 6 h and 18 h. In the thorax group, only four out of six sheep reached a plasma level of 1.5 ng/mL. These sheep achieved the predefined fentanyl plasma level between 9 h and 36 h. At 96 h, no sheep from either group was in the target plasma level range.

For the individual fentanyl plasma concentration peaks see Table 1.

In the foreleg group, one sheep displayed a striking drop in plasma levels after 24 h with values well above the 1.5 ng/mL threshold before (18 h) and after (36 h).

### 3.2. Secondary Outcomes

In all sheep, the general behavior and bodyweight were affected by the TFP. All sheep lost weight during the study. The average weight loss was 3.7 kg (range from 1–9 kg) at 120 h. The average weight loss was 4 kg for the foreleg group and 3.3 kg for the thorax group. The parameters that remained unchanged were respiratory rate (12–40/min), rectal body temperature (38.0–39.5 °C), water uptake and amount and texture of feces, which remained within normal limits for all sheep at all time points. Food uptake was reduced for the first three days for all sheep.

For all sheep, the bandage stayed in place regardless of the location. After TFP removal, the skin was slightly reddened and slightly moist in half of the sheep. These observations were made in both groups. In one sheep of the foreleg group, a part of the bandage rubbed the skin and created a 2 × 0.3 cm superficial wound.

During the first 36 h after TFP application, all sheep showed similar alternating phases, of being apathetic or excited. Apathy was observed as a reduced response to environmental influences (observer, food etc.). Excitement was shown as nervous behavior or pressing their head against the door of the fence when the observer entered the pen. Three sheep were mainly extremely nervous, 3 others head-pressed, and during this time period two sheep were observed frequently walking back and forth along the fence.

Eight out of 12 sheep displayed hypersalivation. These behavioral abnormalities started in six sheep after 3 h, in four sheep after 6 h, and in two after 12 h. In seven sheep, the abnormal behavior disappeared 72 h after TFP application. For the rest after 36 h. These behavioral changes occurred in both groups equally and were not related to a specific fentanyl plasma level.

## 4. Discussions

Aim of this study was to quantify and compare the characteristics of fentanyl uptake following the application of a TFP at two different locations, namely the foreleg and the thorax, in healthy adult sheep. It was hypothesized that the absorption of fentanyl at the thorax is better, leading to a faster onset of action and a higher peak of fentanyl plasma levels than at the foreleg. Contrary to our hypothesis, the faster absorption and highest plasma concentration values were observed for TFP applied on the foreleg. For this patch location, the lowest peak measured was 8.01 ng/mL and five out of six sheep reached fentanyl plasma levels > 1.5 ng/mL after 3 h, whereas for the patch applied to the thorax, the highest concentration reached was 7.87 ng/mL, only one sheep achieving levels above 1.5 ng/mL after 3 h.

It has been shown that the transdermal uptake rate of fentanyl is influenced by the characteristics of the skin. Especially the subcutaneous fat content [29], but also other characteristics of the skin such as skin thickness, corneal layer, temperature, blood vessel architecture, as well as perfusion [30,31] have an impact on absorption in transdermal drug delivery. The specific skin characteristics of the foreleg seemed to favor the fentanyl uptake at this location compared to the thorax. Lyne et al. described significant differences of the skin composition and thickness between sheep breeds and differences depending on the skin location. Skin areas with wool were thinner compared to skin areas with no or less hair [32].

In line with this, a fast and efficient uptake of fentanyl from the foreleg has been previously reported by Ahern et al. in sheep [22]. However, in horses it has been shown that the application of a TFP to the foreleg results in a lower absorption rate and lower fentanyl plasma level when compared to the application to the groin region or thorax [21]. This was confirmed in another study in horses where the TFP applied at the thorax resulted in a fast uptake of fentanyl [22]. The discrepancy between the reported finding in horses and the present study might be due to differences in skin composition between species. According to Riviere et al. [31], differences in the pharmacokinetics of fentanyl uptake between species is linked to differences in the thickness of the stratum corneum. Bouclier et al. described morphological differences between species (rabbit, rat, mice, minipig) and humans [33]. In addition, Bartek et al. showed differences in terms of skin permeability in rat, rabbit, pig, and human [34].

In sheep, other locations for TFP application have also been described in the literature. Jen et al. [35] applied a TFP at the intrascapular region resulting in lower values and a slower uptake than the one achieved in the foreleg group of the present study. In another study, Musk et al. reported a sufficient analgesia applying the TFP at the groin region of pregnant sheep undergoing a hysterotomy and laparotomy. The efficacy of analgesia was assessed via a pain score postoperatively [36]. These two examples as well as the present study highlight differences in fentanyl uptake when patches are applied at different locations in sheep and thereby enabling scientists to use these differences to refine their analgesic protocols. Other possibilities to refine postoperative pain management in sheep are discussed in an article by Lane et al. [37] mentioning a transdermal patch releasing buprenorphine for seven days. A buprenorphine patch was applied in minipigs in a study by Thiede et al., in which a therapeutic plasma levels were reached at least for 72 h [38]. Considering the behavioral changes observed in our study after fentanyl patch application, it could be of interest to compare clinical efficacy and side effects of fentanyl and buprenorphine administered via transdermal route, buprenorphine being potentially associated with lower incidence of side effects.

Behavioral changes (inter alia dysphoria, hypersalivation) were observed after patch application in both groups. These behavioral changes were also described by Marchionatti et al. in a Holstein calve [39]. The TFP dosage of 2 µg/kg/h used in this study is based on the literature [22,26]. In sheep undergoing surgery and treated with the dosage of 2 µg/kg/h, this dysphoria is normally not seen at our institution. In dogs and cats it is described that pain from a surgery can hide side effects like dysphoria due to opioids [40]. Most likely, the “missing” pain in this study conducted in healthy, pain free animals could be a reason for the observed behavior. These side effects were very likely caused by fentanyl since all sheep underwent an uneventful clinical examination by a veterinarian at the beginning of the study. The sheep were normal again after the first 36 h. However, a reliable correlation of the side effects to fentanyl would only be possible with a control group undergoing the same procedure (sedation, long-term catheter, bandage, blood taking) without a TFP application.

Hypersalivation caused by opioids is to our knowledge not described in sheep. However nausea and vomiting are described in small animals [14] and might have caused hypersalivation in the sheep of the present study. In addition, hypersalivation is described in dogs after application of a TFP [41]. The sheep in this study lost weight most likely due to the behavioral changes in the first days leading to reduced food uptake. This weight loss as an additional side effect indicates again the need of an adequate dosage of fentanyl adapted to the surgical procedure. No other side effects of fentanyl described in literature (i.e., respiratory depression, reduced food uptake, reduced gut motility, and hyperkinesia [14,42]) were observed in this study

From a practical point of view, the application of the TFP to the foreleg was slightly easier. However, the risk of tightening the bandage too much and creating a wound or edema, is higher at the foreleg. Regular controls of the bandage after TFP application are highly recommended.

The major limitation of this study is assuming the same fentanyl plasma level being analgesic in sheep as in humans. To our knowledge, no study identifying the minimal analgesic level in sheep has been published and even for humans a wide range is reported in the literature. Frequently referenced by other veterinary studies [38,39,43] is the range between 0.5 and 2 ng/mL as the minimal analgesic level published in a review in the early nineties [44]. In an older study, 0.6 ng/mL has been reported [45]. In a more recent article by Grape et al., the analgesic plasma level for humans of 0.6–1.5 ng/mL was described and used as a reference in this study [28]. Even though it is common practice to apply the human minimal analgesic level for different species in veterinary medicine, it would clearly be beneficial to establish these levels for frequently treated species. However, as it is already challenging to define the minimal analgesic level in humans, it is even more so in animals. Therefore, the present study was designed without surgical intervention or other painful stimuli to avoid interaction with the additionally necessary medication and surgery as well as individual differences in pain sensitivity between animals. Thus, evaluation of the analgesic effectiveness using a pain score was not feasible. Further studies are needed to describe the minimum analgesic level for fentanyl in sheep, using quantitative assessment methods like for example the nociceptive withdrawal reflex [46].

Fentanyl plasma levels can be measured by different methods, which may result in some discrepancies regarding the reported levels between studies. In this study a commercial ELISA-Kit was used. Ahern et al. [22] analyzed samples using liquid and gas chromatography mass spectrometry. Even though the same dose was applied to the animals, the measured fentanyl plasma levels were substantially lower compared to the present study. Using an ELISA-Kit from a different company, Christou et al. [20] measured lower values as well. However, Jen et al. [35] showed that both methods mentioned are positively correlated. Therefore, the difference between the two locations described here is valid even though the peak values may differ if a different detection method would have been used. In the foreleg group, one sheep displayed a striking drop in plasma levels after 24 h. This sample might be diluted due to unknown reasons. No other explanation can be given.

Only female sheep were used in this study, as it was shown in a recent European survey that female sheep are used significantly more often in research [1,4,5,6,7,8,9,47] most likely due to the fact that males are usually castrated shortly after birth. However, Solassol et al. showed that sex does not influence fentanyl uptake in humans [48] and to our knowledge, information about the effect of sex on transdermal fentanyl uptake in sheep is not recorded. Suárez-Morales et al. compared fentanyl doses needed by male and female human patients undergoing general anesthesia and showed a higher dose level was required in females [49]. Many other studies investigating the influence of sex on the response to opioids in rodents [50] showed the same tendency with males having a higher sensitivity to opioids [51,52,53]. However, this result could not be shown in a study using fentanyl in Sprague-Dawley rats [54].

In an experimental animal study, the analgesia protocol should provide reliable and appropriate pain relief in all sheep, and therefore decreasing the need for additional, individual pain medication which could impact the outcome. In this study, the TFP applied to the foreleg demonstrated a reliable onset of fentanyl uptake in all animals. After 6 h, all sheep of the foreleg group reached the level of 1.5 ng/mL fentanyl plasma level and five of these after 3 h. Therefore, an application of the TFP 12 h before the desired onset of effect as done by Ahern [22] may not be needed.

## 5. Conclusions

A TFP, applied either on the foreleg or on the thorax, induced an effective fentanyl uptake in healthy sheep. However, a faster uptake, longer duration of action and easier application were observed for the foreleg compared to the thorax. Assuming that the effective plasma concentration described for humans is providing analgesia in sheep as well, the present findings suggest that it should be sufficient to apply the TFP 3–6 h before the painful insult and that its effect should last at least 48 h.

## Figures and Tables

**Figure 1 animals-10-01675-f001:**
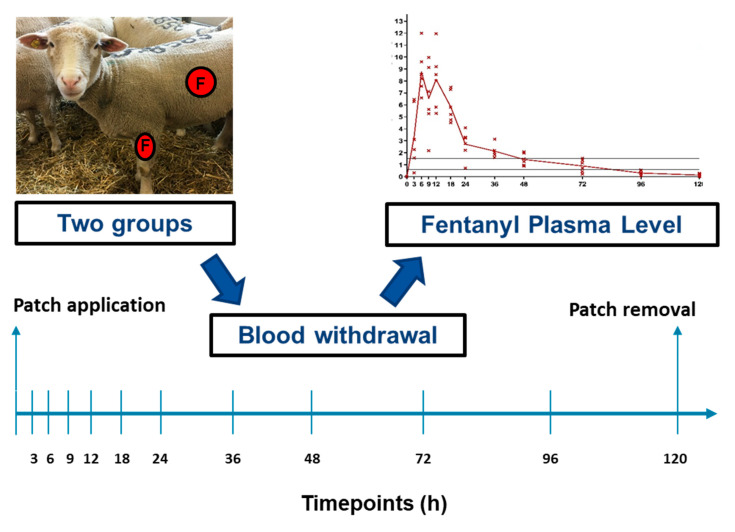
Study Design Overview—The letter “F” in the first picture shows the locations of the TFP on the sheep. However, only one location was used per sheep. After application of a transdermal fentanyl patch (TFP) to one of two possible locations depending on group allocation, fentanyl plasma levels were measured at different time points.

**Figure 2 animals-10-01675-f002:**
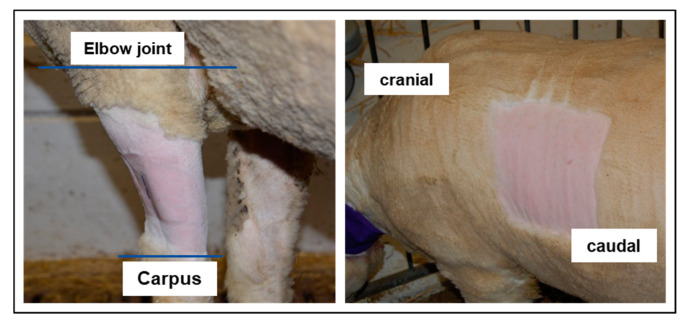
Transdermal fentanyl patch location—left picture: foreleg, right picture: thorax.

**Figure 3 animals-10-01675-f003:**
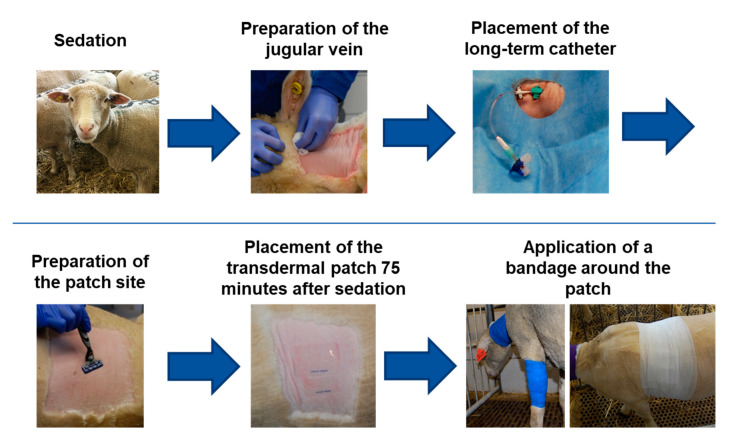
Work flow of the long-term catheter placement and transdermal fentanyl patch application.

**Figure 4 animals-10-01675-f004:**
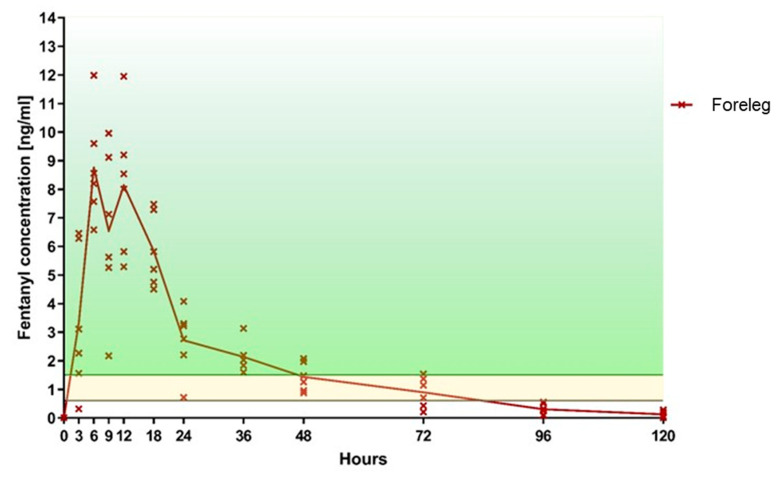
Foreleg—Fentanyl plasma level over time (the fentanyl level within the area marked in yellow, correspond to the minimal analgesic plasma level of 0.6–1.5 ng/mL described for humans; the area above marked in green represents values higher than 1.5 ng/mL).

**Figure 5 animals-10-01675-f005:**
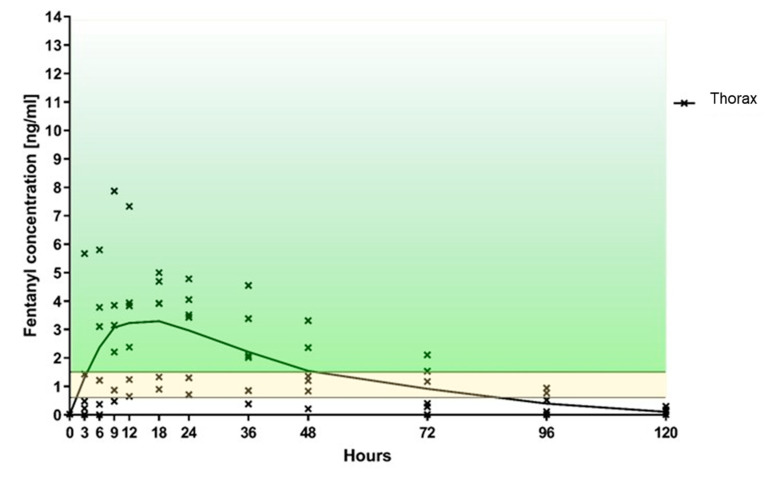
Thorax—Fentanyl plasma level over time (the fentanyl level within the area marked in yellow, correspond to the minimal analgesic plasma level of 0.6–1.5 ng/mL described for humans; the area above marked in green represents values higher than 1.5 ng/mL).

**Figure 6 animals-10-01675-f006:**
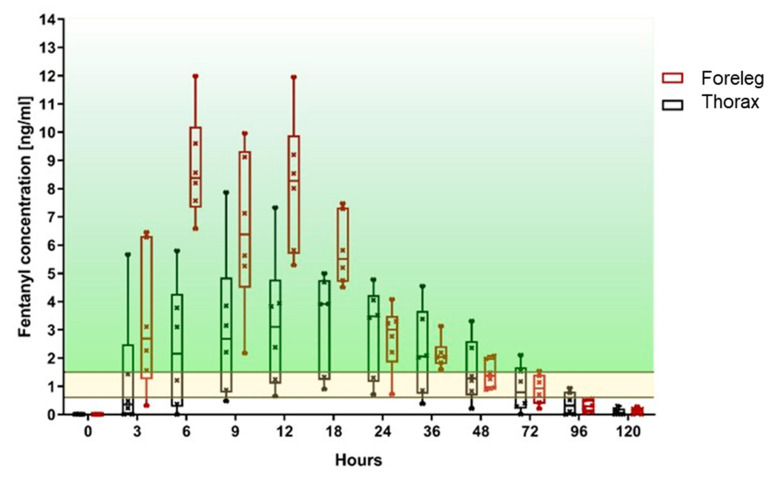
Fentanyl plasma levels measured over time for TFP applied at the foreleg (**red**) and thorax (**black**) displayed in a box plot graph (the fentanyl level within the area marked in yellow, corresponds to the minimal analgesic plasma level of 0.6–1.5 ng/mL described for humans; the area above marked in green represents values higher than 1.5 ng/mL).

**Table 1 animals-10-01675-t001:** Fentanyl plasma level of both groups (values marked in yellow are above 0.6 ng/mL; values marked with green are above 1.5 ng/mL; every column represents one sheep).

	Fentanyl Concentration ng/ml											
	Foreleg						Thorax					
**Timepoint**	**1**	**2**	**3**	**4**	**5**	**6**	**7**	**8**	**9**	**10**	**11**	**12**
**0 h**	0.01	0.01	0.01	0.01	0.01	0.01	0.02	0.01	0.01	0.01	0.01	0.01
**3 h**	3.11	6.46	1.56	0.32	2.27	6.28	5.67	0.01	1.43	0.01	0.48	0.23
**6 h**	8.56	7.57	6.58	8.20	11.99	9.60	5.80	0.37	3.78	0.00	3.10	1.21
**9 h**	9.12	9.96	5.63	7.13	5.26	2.17	7.87	0.48	3.85	0.87	3.15	2.21
**12 h**	9.20	5.29	8.54	8.01	11.95	5.82	7.33	0.65	3.94	1.24	3.83	2.38
**18 h**	7.29	4.51	4.75	5.20	7.48	5.82	5.00	0.90	3.91	1.33	3.92	4.69
**24 h**	2.21	3.30	4.08	0.72	2.77	3.23	4.05	0.71	3.52	1.30	3.43	4.78
**36 h**	2.03	1.84	3.13	1.60	2.19	2.04	2.09	0.38	3.38	0.85	2.02	4.55
**48 h**	1.26	0.88	1.97	1.48	0.96	2.08	1.20	0.21	2.36	0.83	1.36	3.31
**72 h**	1.37	0.71	1.14	0.43	0.21	1.54	1.17	0.01	1.53	0.40	0.28	2.11
**96 h**	0.51	0.08	0.32	0.56	0.11	0.26	0.52	0.01	0.77	0.11	0.01	0.94
**120 h**	0.19	0.01	0.20	0.07	0.01	0.28	0.30	0.01	0.18	0.01	0.01	0.11

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
