# Peer review of "Transdermal Fentanyl Uptake at Two Different Patch Locations in Swiss White Alpine Sheep"

_animals, 2020, doi:10.3390/ani10091675_

Round 1

Reviewer 1 Report

Your work presents merit and is particularly important because aimed to improve the knowledge of pain control, which is a very important task.

Please attend my commentaries below that I hope to increase the general quality of your manuscript.

The breed of sheep used shall appear in the title

Keywords: should include “sheep” or “sheep analgesia”

Line 67 - “In addition, pain assessment during the postoperative phase is very challenging, and insufficient analgesia could easily be unrecognized. “ – Should be referenced, many works reveal the difficulty of pain evaluation in this and similar species.

Material and Methods –How were the animals randomized? Using the earring number, ruminal chip number, or another method? Please explain if the animals were previously submitted to surgeries that used similar anesthetics as the pharmacokinetics of drugs change after consecutive administrations. The groups were of the same size? I know that this is referred in results, but should be referred to in the material.

The breed of sheep are all the same? Do you measure the skin thickness? These results can be transposed to other breeds of sheep that may have different skin thicknesses? What about goats? Please put this data in the methods or in the discussion.

Line 125 – What is “bloodwork”? I don't recognize that term. Please replace for “clinical examination, hemogram (hematocrit and white blood cell count) and biochemistry analysis (total proteins). Please refer to the machines used for analysis. They are indicated for this species analysis? If not, you included a control analysis?

Why you just used females? Please explain. The males will have the same response? Refer if exists bibliography that evaluated fentanyl pharmacodynamics and pharmacokinetics in male and females.

Figure 3 - You used the Gillette to prepare the patch site because the sheep were shaved. What if they don't? Do you just use the Gilette? Of course, you'll have to shear in another way first. Please give this data to readers. A Gillette was used for how many animals? If for several, how did you disinfect it among animals?

Line 176 – “As an analgesic fentanyl plasma level for sheep is not described in the literature, the minimum analgesic fentanyl plasma level  described for opioid-naive human patients …” This is not true and is a serious statement that should not happen, since it is very easy to find other works that extrapolated and calculated reference values, including for different breeds of sheep. Please see for example and review it “Pharmacokinetics of a Transdermal Fentanyl Solution in Suffolk Sheep (Ovis aries)”, “Comparison of the analgesic properties of transdermally administered fentanyl and intramuscularly administered buprenorphine during and following experimental orthopedic surgery in sheep”, “Pharmacokinetics of fentanyl administered transdermally and intravenously in sheep”, “Transdermal fentanyl and its use in ovine surgery” and many others.

Line 251 “In all sheep, the general behavior and bodyweight were affected by the TFP. All sheep lost 251 weight during the study.” The study didn’t include controls, such as, a group of animals that were submitted to anesthesia, catheterization, and colocation of bandage but not fentanyl patch. This is a critic design problem, not compatible with the "Refinement" that the job begins by talking about. In fact, we do not know whether weight loss and other animal welfare changes are due to the process of anesthesia, catheterization, bandage placement, patch placement, or others.

According to the previous bibliography, the MEC of fentanyl in plasma in sheep was determined to be 0.5 to 2 ng/Ml by different authors. Why your value is quite different? Why you don’t use this value? Please explain. If you don’t have an explanation, please reformulate your cut off value.

Author Response

Dear reviewer

Thank you for your constructive comments to our manuscript. We believe that the necessary changes improved our manuscript substantially.

Please find below our detailed replies point to point.

The respective line specifications for the manuscript are valid for the review setting "show all markup".

Best regards

Stephan Zeiter

Reviewer 2 Report

This paper is a useful addition to the data on the use of analgesia in sheep after surgery.   It compares plasma levels of fentanyl after transdermal patch applications at 2 sites.  Overall it is well written and clear, but I cannot reconcile the text with the corresponding table.  The Table and the text figures correspond and are clear, apart from some nomenclature.  They have also made some interesting observations of the behaviour of the animals after application that I feel need to be elaborated as they are important.

I have attached a rather poor 'pdf to docx' conversion of the text with tracked comments and changes to try to help the authors with English usage and idioms.

My overall remarks and corresponding approximate page numbers, and other major comments are given below.  Other minor corrections will be found in the accompanying docx file.

Line 16: All locations are 'anatomical' in some way.  In this case the key issue is that it is a superficial location?

Line 21 and elsewhere: Animals have ‘legs’ and not ‘arms’, so it should be ‘forelegs’ everywhere.

Line 55: The key point is that it is an intramedullary pin (as stated in the original paper.  It is not a human humerus pin.

Figure 1:  Withdrawal is misspelt.

Line 121: I have reassorted the key details of the animals – please check

Lines 213-219: These figures do not seem to reflect those in Table 1.  Please check.  The text figures however, do correlate well with Table 1.  I am confused by the text and Table 1.

Table1, Figures 4 and 6: Correct forearm to foreleg

Lines 261-266: These behaviours are all very different (e.g. excitement and apathy) but they are important observations.  Head pressing can be a sign of headaches, and restlessness could be pain/distress related or some other mental disturbance.  Hypersalivation is an unusual sign.  I suggest you consult a sheep vet or ethologist in your institute and then expand and rewrite the paragraph and include some further comments in the discussion..

Author Response

(The authors gave the same response as above.)

Round 2

Reviewer 1 Report

Dear colleagues,

Thanks for the quick and pointed review of some content. I also believe that the work is now clearer and thank you for the doubts that have cleared me up.

Kind regards

Author Response

Dear reviewer

thank you for your valuable input improving our manuscript.

Best regards

Reviewer 2 Report

This is a much improved version but I still have some concerns over wording which I have tried to help.

Please note that patients explain their symptoms to their doctors, but doctors observe the clinical signs in their patients.  So vets and behaviourists observe signs in the animals, not symptoms.

Line 109- 110: the phrase is ‘drawing lots’.  Change to “The sheep were randomly assigned to one of two equal groups (n=6 per group) by drawing lots.”

Line 162: Sheep included in this study were all female and more than 2 years old,

Line 165 None of the sheep had undergone anaesthesia or surgery to exclude any prior effects.

Lines 246-259:  I am still confused by these lines and part of that confusion is the change in units, so I have made the suggested wording below. 

Based on the selected TFP size, the overall average expected fentanyl administration rate was 2080ng/h/kg (range 2010 to 2160ng/h/kg). The average expected fentanyl administration rate was 2010ng/h/kg (range 2010 to 2160ng/h/kg) in the foreleg group and 2070ng/h/kg (range 2010 to 2100ng/h/kg) in the thorax group.

In the foreleg group, 5 out of 6 sheep achieved a fentanyl plasma level threshold  of >0.6ng/ml after 3h, and all sheep within 6h.  At least this level was maintained in all sheep for a minimum of 48h and in 4 out of 6 sheep for a maximum of 72h (Table 1, Figure 4).  In the thorax group, only 2 sheep had fentanyl levels above 0.6 ng/ml after 3h, 4 sheep after 6h, 5 sheep after 9h, and in all six sheep after 12h (Table 1, Figure 5).  At 24h, all six sheep still had a concentration above 0.6 ng/ml.  However, at 48h it started to decline so that by 72h the level of 0.6ng/ml was achieved in only 5 sheep, at 72h in 3 sheep, and by 96h no sheep had a level above 0.6ng/ml.  At 120h, no sheep from either group was above 0.6ng/ml.  When comparing both groups, the fentanyl uptake was more consistent in terms of the onset of the desired plasma concentration as well as duration of effect (Figure 6), when applied to the foreleg than to the thorax.

Lines 302-309: I suggest replacing this text with “During the first 36 hours after TFP application, all sheep showed similar alternating phases of being apathetic or excited.  Apathy was observed as a reduced response to environmental influences (observer, food etc.).  Excitement was shown as nervous behaviour or pressing their head against the door or the fence when the observer entered the pen. Three sheep were mainly extremely nervous, 3 others head-pressed, and during this time period two sheep were observed frequently walking back and forth along the fence.”

Line 342: should read “…rat, rabbit, pig and human [35]

Lines 407 -416:  I suggest to replace with:  “Only female sheep were used in this study, as it was shown in a European survey that female sheep are used significantly more often in research [1, 4-9, 47] most likely due to the fact that males are usually castrated shortly after birth.  However, Solassol et al. showed that gender does not influence fentanyl uptake in humans [48] and to our knowledge information about the effect of gender on transdermal fentanyl uptake in sheep is not recorded.  Suárez-Morales et al. compared fentanyl doses needed by male and female human patients undergoing general anesthesia [49] and showed a higher dose level was required in females [49]. Many other studies investigated the influence of gender on the response to opioids in rodents[50] showed the same tendency with males having a higher sensitivity to opioids [51-53].  However, this result could not be shown when using fentanyl in Sprague-Dawley rats [54].”

Author Response

Dear Reviewer

thank you for your valuable input - please see our answers in the attached point by point answer.

Best regards
